# Biomechanical Role of Epsin in Influenza A Virus Entry

**DOI:** 10.3390/membranes12090859

**Published:** 2022-09-05

**Authors:** Jophin G. Joseph, Rajat Mudgal, Shan-Shan Lin, Akira Ono, Allen P. Liu

**Affiliations:** 1Department of Mechanical Engineering, University of Michigan, Ann Arbor, MI 48109, USA; 2Department of Microbiology and Immunology, University of Michigan, Ann Arbor, MI 48109, USA; 3Department of Biomedical Engineering, University of Michigan, Ann Arbor, MI 48109, USA; 4Cellular and Molecular Biology Program, University of Michigan, Ann Arbor, MI 48109, USA; 5Department of Biophysics, University of Michigan, Ann Arbor, MI 48109, USA

**Keywords:** clathrin-mediated endocytosis, influenza A virus endocytosis, epsin, live-cell imaging

## Abstract

Influenza A virus (IAV) utilizes clathrin-mediated endocytosis for cellular entry. Membrane-bending protein epsin is a cargo-specific adaptor for IAV entry. Epsin interacts with ubiquitinated surface receptors bound to IAVs via its ubiquitin interacting motifs (UIMs). Recently, epsin was shown to have membrane tension sensitivity via its amphiphilic H_0_ helix. We hypothesize this feature is important as IAV membrane binding would bend the membrane and clinical isolates of IAVs contain filamentous IAVs that may involve more membrane bending. However, it is not known if IAV internalization might also depend on epsin’s H_0_ helix. We found that CALM, a structurally similar protein to epsin lacking UIMs shows weaker recruitment to IAV-containing clathrin-coated structures (CCSs) compared to epsin. Removal of the ENTH domain of epsin containing the N-terminus H_0_ helix, which detects changes in membrane curvature and membrane tension, or mutations in the ENTH domain preventing the formation of H_0_ helix reduce the ability of epsin to be recruited to IAV-containing CCSs, thereby reducing the internalization of spherical IAVs. However, internalization of IAVs competent in filamentous particle formation is not affected by the inhibition of H_0_ helix formation in the ENTH domain of epsin. Together, these findings support the hypothesis that epsin plays a biomechanical role in IAV entry.

## 1. Introduction

Clathrin-mediated endocytosis (CME) plays an essential role in signal transduction and nutrient uptake in eukaryotic cells [1]. CME is the most frequent mode of entry for small and medium-size viruses such as influenza A virus (IAV) and adenovirus [2]. It is known that viruses like SARS-CoV2, HIV and Dengue virus use CME as one of their modes of cellular entry [3,4,5,6,7]. Viral entry by CME is an energetically costly biomechanical process involving plasma membrane deformation and clathrin-coat assembly around the viral particle [8,9,10]. The shape, size, stiffness, and other physical characteristics of the particle control the energetic requirement of this process [8,10,11,12]. However, the mechanisms that drive the internalization of these viral particles and how the physical properties of the virus influence the internalization via receptor-mediated endocytosis remain elusive. Better understanding of the biomechanics of viral entry is necessary not only in developing better therapeutics against infectious diseases, but also in designing viral vector and nanoparticle-based drug delivery systems [13,14,15,16].

IAVs are considered a major cause for annual epidemic and occasional pandemic due to their more rapid antigenic drift [17,18]. Internalization of IAVs is mediated through multiple endocytic pathways [19]. It has long been identified that influenza virions utilize CME as their endocytic route for their entry into cells upon binding of viral hemagglutinin (HA) glycoproteins to surface sialic acids [19,20]. Evidence obtained from live-cell imaging has revealed the formation of clathrin-coated pits (CCPs) at viral binding sites and the requirement for epsin 1 (hereafter referred to as epsin), a CCP adaptor protein, during clathrin-mediated entry of IAVs [21]. During IAV entry, ubiquitin interacting motifs (UIMs) in epsin bind to ubiquitinated cell surface receptors at the site of IAV attachment, inducing internalization of IAV.

Recent findings demonstrate that the H_0_ helix in the N-terminus of epsin can act as a membrane tension sensor, and that epsin promotes and stabilizes clathrin coat assembly at high membrane tension conditions [22]. It has also been shown that H_0_ helix in ENTH/ANTH proteins can detect and generate membrane curvature [23,24], pointing out a possibility that in addition to being a cargo-specific adaptor for IAVs, epsin also plays a biomechanical role in curvature and clathrin coat assembly and stabilization during viral entry. Since clinical isolates of IAVs appear in both filamentous and spherical shapes, and endocytic pathways mediating IAV entry may depend on virion morphology [25,26,27], we hypothesize the membrane tension sensing ability of epsin plays a role in CME-mediated IAV uptake. Here, by using full-length epsin and epsin mutants without functional H_0_ helix as well as IAVs of different morphologies, we revealed the biomechanical significance of epsin during IAV entry. By using live cell lattice light sheet microscopy, total internal reflection fluorescence (TIRF) microscopy and 3D-SIM (structured illumination microscopy), we captured the viral entry and analyzed the colocalization of epsin and its mutants with different strains of IAVs. We further used flow cytometry to measure the bulk uptake of different strains of IAVs in cells expressing epsin mutants. Altogether, we found that the curvature sensing ability of epsin H_0_ helix, by detecting the local changes in membrane curvature upon viral binding, is critical for the entry of IAVs of different morphology via CME. We also demonstrated that the ENTH domain is crucial for the ability of epsin to act as a cargo-specific adaptor for IAV entry.

## 2. Materials and Methods

### 2.1. Cells and Reagents

Stable RPE cells overexpressing WT epsin or expressing epsin-EGFP or epsin-EGFP mutants and mCherry-clathrin light chain (Clc) were generated as previously described in Joseph et al. [22]. Epsin mutants were generated using Q5^®^ Site-Directed Mutagenesis Kit (New England Biolabs Inc., Ipswich, MA, USA). ENTH domain in epsin-EGFP was removed by mutagenesis to generate epsin ΔENTH EGFP using forward primer 5′-GCCCACGCGCTCAAGACC-3′; reverse primer 5′-CATGGTGGCCTCGAGATCTGAG-3′. 18 amino acids on the N-terminus of epsin which form H_0_ helix was replaced by 6 histidine to generate epsin mut-H_0_ EGFP using forward primer 5′-CACCACCACAACTACTCAGAGGCAGAGATC-3′ and reverse primer 5′-ATGATGATGTGTCGACATGGTGGCCTC-3′. UIM motifs were removed from epsin via mutagenesis using forward primer 5′-GAGGAGTCATCTCTTATGGATCTTGCTGAC-3′ and 5′-CCCGCTGCTCTGTGGCCA-3′.

Stable cells expressing EGFP-Clc were transiently transfected with CALM-mCherry (Addgene, Watertown, Massachusetts, USA). For transfection, cells at 70% confluency were transfected with the desired DNA constructs using lipofectamine 2000 (Invitrogen, Waltham, MA, USA, #11668019) in Opti-MEM (Gibco, Thermo Fisher Scientific, Waltham, MA, USA, #31985062). These cells were cultured in Dulbecco’s Modified Eagle Medium with nutrient mixture F-12 (DMEM/F12) supplemented with 10% (*v/v*) fetal bovine serum (FBS) and 2.5% (*v/v*) penicillin/streptomycin at 37 °C and 5% CO_2_.

### 2.2. Generation of Influenza A Virus Strains with Different Morphologies

Influenza A virus (H3N2: strain A/Aichi/2/68) was purchased from ATCC. The stocks of influenza A viruses A/WSN/1933 (H1N1) and its derivative viruses were generated using a reverse genetics system [28] by transfecting HEK293T cells with 12 plasmids encoding genome segments of influenza virus and viral proteins needed for genome replication, followed by further propagation in MDCK culture. Viruses of different morphology were generated by replacing the M segment encoding M1 and M2 structural proteins of IAV [29]. WSN WT strain is known to produce spherical IAV morphology. WSN-UdM strain which forms filamentous IAVs were generated by replacing M segment of WSN-WT with M segment of A/udorn/1972 strain (H3N2) [29] (Appendix A). Isogenic control (WSN-UdM1A) of WSN-UdM was generated by a point mutation in M segment of WSN-UdM which abrogates the ability of producing filamentous particles. Plaque forming units (PFU), which show the viral titer of IAV variants were determined using a plaque assay as previously performed [30].

### 2.3. Fluorescence Labeling of IAV

IAVs were labeled with lipophilic dye DiD (Invitrogen, Waltham, MA, USA, #V22887). 100 µL original virus stocks of each strain were incubated with 1 µL of 1 mM DiD dissolved in DMSO at room temperature with gentle vortexing. Unincorporated dye was removed by centrifugation at 5000 *g* followed by media exchange using Amicon Ultra-15 centrifugal filter units (Millipore Sigma, St Louis, MO, USA, #UFC903096). Labeled IAVs were visualized using epi-fluorescence microscopy (Appendix A).

### 2.4. Inhibiting Clathrin-Mediated Endocytosis

RPE cells stably expressing epsin-EGFP or epsin mut-H_0_ EGFP and mCherry-Clc were incubated with Pitstop 2 (20 µM) (Abcam) for 15 min at 37 ºC for inhibiting CME prior to influenza viral infection.

### 2.5. TIRF Microscopy

RPE cells stably expressing epsin-EGFP or epsin-EGFP mutants and mCherry-Clc or transiently expressing CALM-mCherry and EGFP-Clc were seeded on glass-bottom dishes (#1.5, Mat-tek Corp.) 12 to 16 h prior to experiment. DiD-tagged IAVs (7 – 8 × 10^6^ PFU/mL) were added to the dishes and incubated at 37 °C for 1 h. Immediately after, the cells were washed with PBS and fixed using 4% paraformaldehyde for 10 min on ice. TIRF microscopy was performed to image the colocalization of IAVs at the basal layer of cells using a Nikon TiE-Perfect Focus System (PFS) microscope equipped with an Apochromat 100X objective (NA 1.49), a sCMOS camera (Flash 4.0; Hamamatsu Photonics, Shizuoka, Japan), and a laser launch controlled by an acousto-optic tunable filter (AOTF). Fixed cells were imaged at 100 ms exposure with excitation of 488 nm (EGFP), 561 nm (mCherry), 640 nm (DiD) lasers (Coherent Sapphire, Santa Clara, CA, USA).

### 2.6. D Structured Illumination Microscopy

RPE cells stably expressing epsin-EGFP or epsin mut-H_0_ EGFP and mCherry-Clc were seeded on 8-well chamber culture slide (#1.5, Nunc Lab-Tek) 12 to 16 h prior to each experiment. Cells were infected with labelled WSN-WT and WSN-UdM IAVs for 1 h. Immediately after, the cells were washed with PBS and fixed using 4% paraformaldehyde for 10 min on ice. 3D-SIM was performed using a Nikon N-SIM microscope equipped with an Apochromat 100X objective (NA 1.49) and a sCMOS camera (Flash 4.0; Hamamatsu Photonics, Shizuoka, Japan). Images were acquired in 3D SIM mode with nine images taken at each z depth of 200 nm of each color with linear translation of Moire pattern for SIM reconstruction. EGFP-epsin, mCherry-Clc and DiD-labeled IAVs were acquired at each z-step with 488 nm, 561 nm, and 640 nm excitation at exposure times of 100 ms, 100 ms and 50 ms respectively.

### 2.7. Live-Cell Lattice Light Sheet Microscopy

RPE cells stably expressing epsin-EGFP or epsin mut-H_0_ EGFP and mCherry-Clc were seeded on 8-well chamber culture slide (#1.5, Nunc Lab-Tek) 12 to 16 h prior to each experiment. DiD-tagged IAVs (7–8 × 10^6^ PFU/mL) were added to each well and immediately mounted on the microscope. Viral entry to the whole cell volume was imaged for a duration of 30 min using ZEISS (Carl Zeiss Inc., Oberkochen, Germany) lattice lightsheet 7 (44.83X/1.0 NA Objective at 60° angle to the cover glass, Pco.edge 4.2 CLHS sCMOS camera). The whole volume of a cell was imaged in 30 s with an acquisition sequence of 15 ms exposure for 488 nm, 561 nm and 640 nm excitations at individual imaging depth. Lightsheet images were deskewed, deconvolved, drift corrected and transformed to cover glass orientation using Zeiss Zen 3.5 software (Carl Zeiss Inc., Oberkochen, Germany). Deskewed, transformed image in cover glass orientation has a voxel size of 145 nm × 145 nm × 145 nm.

### 2.8. Colocalization Analysis for Fixed Cells

IAVs and CCS puncta with proteins of interest were detected by performing Gaussian mixture model fitting using custom–written software in MATLAB (R2013b, The MathWorks Inc., Natick, MA, USA) as previously described in Aguet et al. [31]. Percentages of IAVs colocalized with epsin, epsin mutants, CALM and Clc were calculated. If the detection mask of individual IAVs overlapped with detection masks of protein puncta, such IAVs were counted as colocalized with the corresponding protein. Further, the strength of recruitment of proteins to IAV-containing CCSs was determined. Around each detected IAV colocalized with proteins of interest, an annulus region was considered with an outer-radius two times and inner-radius equal to the radius of the IAV. Ratio of average intensity of proteins in the IAV puncta and in the annulus region around IAV was calculated. Proteins were considered preferentially recruited to the IAV puncta if the intesity ratio was greater than 1. The intensity ratio of the epsin, epsin mutants, CALM and Clc were compared to determine the relative strength of recruitment.

### 2.9. IAV Particle Tracking for Lattice Light Sheet Microscopy

Individual trajectories of IAVs were detected and tracked using TrackMate plugin in Fiji (ImageJ) [32] from the deskewed and transformed 3D time lapse images of viral entry. A detector with Laplacian of Gaussian filter is applied to detect IAVs with a quadratic fitting scheme for subpixel localization and estimated blob size parameter of 3 µm. A simple Linear Assignment Problem (LAP) tracker was applied with a maximum linking and gap closing distance of 3 µm and maximum gap closing of 2 frames. Using a custom written software in MATLAB, time-lapse montage of IAV particle images (3 × 3 µm) overlayed with corresponding epsin-EGFP and mCherry-Clc images were generated by utilizing the spatial and time coordinate of IAVs tracked by TrackMate. Subsequently, using edge detection, epsin and clathrin puncta in the montage co-localizing with the IAV tracks were detected. An IAV track was considered colocalized with epsin or clathrin if it had overlapping epsin or clathrin puncta in more than three consecutive frames. IAV tracks were classified into three populations: (i) IAV tracks which are clathrin and epsin positive, (ii) IAV tracks which are clathrin positive and epsin negative, (iii) IAV tracks which are clathrin and epsin negative.

### 2.10. Flow Cytometry

RPE cells expressing different epsin constructs were plated in 24 well plates 12 to 16 h prior to each experiment. Cells were infected with DiD-tagged IAVs (7–8 × 10^6^ PFU/mL) at 37 °C for a time course from 1 h to 4 h. Immediately after, the cells were washed with PBS and lifted with trypsin EDTA (0.25%, Gibco, Thermo Fisher Scientific, Waltham, MA, USA, #25200056), and resuspended in 4% paraformaldehyde fixing solution for 10 min on ice. 50,000 cells were analyzed by Guava EasyCyte Flow cytometer (Milipore Corporation, Hayward, CA, USA) for IAV uptake and each experiment is repeated 3 times. Flow cytometry data was analyzed using Flowing Software (Turku Bioscience Centre, Turku, Finland).

### 2.11. Infectious Virus Measurement by Plaque Assay

MDCK cells were seeded 24 h before infection. Cells were washed once with MEM-BSA and serially diluted IAVs samples labeled with or without lipophilic dye DiD (Invitrogen, Waltham, MA, USA, #V22887) were added to the confluent monolayer of MDCK cells in MEM-BSA and incubated for 1 h at 37 °C. After viral adsorption, cells were washed once and layered with MEM-BSA containing 0.76 µg/mL TPCK-treated trypsin (Worthington Biochemical, Lakewood, NJ, USA) and 1% Seakem GTG agarose /Seaplaque (Lonza, Basel, Switzerland). After the agarose plug solidified, the plates were inverted and incubated at 37 °C. After 48–72 h, the agarose plug was removed, and the cells were stained with 0.1% crystal violet in 20% methanol for 10 min. Plaques were quantified and recorded. 

### 2.12. Statistics and Reproducibility

For intensity ratio data, the statistical significance was verified by one-way ANOVA test followed by post hoc Bonferroni and Holm’s multiple comparison test to determine the *p* values between individual pairs of data. *, **, and *** were assigned to *p* < 0.05, *p* < 0.01, and *p* < 0.001, respectively. For co-localization ratio data from microscopy and IAV-positive cell ratio from flow cytometry, the statistical significance between individual pairs of data was verified using binomial test. *, **, and *** were assigned to *p* < 0.05, *p* < 0.01, and *p* < 0.001, respectively. Each experimental condition was repeated at least three times for flow cytometry and microscopy with data collected from multiple cells. The number of cells considered for each analysis is provided in the corresponding figure captions.

## 3. Results

### 3.1. IAVs Co-Localize with Clathrin-Coated Structures (CCSs) Containing ENTH/ANTH Proteins

IAV is known to enter cells via CME by hijacking the activity of endocytic proteins [33,34]. ENTH/ANTH proteins play an important role in membrane curvature formation during CME [35,36]. To study how ENTH/ANTH proteins are involved in IAV entry, we utilized stable retinal pigment epithelial (RPE) cells expressing epsin-EGFP, an ENTH family protein, or transiently expressing clathrin assembly lymphoid myeloid leukemia protein (CALM)-mCherry, an ANTH family protein (Figure 1a). Previous work from our lab has shown that RPE cells have very low endogenous expression of epsin, making the cell line an attractive platform to perform studies with over expression of epsin-mutants by minimizing the effect of endogenous epsin [22]. Both epsin and CALM have N-terminal structured membrane-binding domains and C-terminal unstructured domains with multiple binding motifs to other endocytic proteins. The RPE clones stably expressed mCherry-clathrin light chain (Clc) or EGFP-Clc, respectively, and allowed live cell tracking of epsin-containing or CALM-containing CCSs. These cells were infected with DiD-labelled IAVs (H3N2: strain A/Aichi/2/68) and imaged using TIRF microscopy to determine colocalization of IAVs with epsin-EGFP or CALM-mCherry (Figure 1b).

We found that ~90% of IAVs localizing in CCSs showed enrichment of epsin (Figure 1c), while cells expressing CALM-mCherry had a reduced enrichment of ~65%, suggesting that UIM domain might be important for recruitment of CME adaptor proteins to IAV binding sites. Even though CALM is structurally similar to epsin, it does not possess UIMs which are shown to interact with ubiquitinated surface receptors bound to IAVs [21,22,23,24,25,26,27,33,34,35,36]. We also analyzed the strength of recruitment of epsin or CALM to IAVs by quantifying the ratio of intensity of protein puncta (i.e., epsin or CALM) over the background (i.e., outside of puncta) protein intensity in CCSs colocalized with an IAV (Figure 1d, see Methods for more details). This ratiometric quantification showed higher recruitment of epsin compared to CALM to CCSs at IAV binding sites. Both results confirm earlier findings that epsin is a cargo-specific adapter for CME of IAVs [21].

### 3.2. Colocalization of IAVs to CCSs and Internalization Are Disrupted by Overexpression of Epsin Mutants without Functional ENTH Domain

ENTH domain of epsin is shown to be biomechanically active [22]. The H_0_ helix in the N-terminus of ENTH domain acts as a tension sensor and can detect changes in area per lipid in the lipid bilayer. It has been shown that overexpression of epsin did not affect the bulk uptake of IAV in RPE cells and the removal of UIMs from epsin can reduce the colocalization of IAVs with CCSs [21]. Building on these findings, we investigated whether removal of biomechanically active ENTH domain also elicited a similar disruption of IAV colocalization with epsin-containing CCSs.

To investigate the requirement of ENTH domain during IAV entry, we first examined the colocalization of IAVs with CCSs enriched with different epsin mutants that have been previously characterized and used [22]. RPE cells stably expressing epsin ΔENTH EGFP, epsin mut-H_0_ EGFP, epsin ΔENTH ΔUIM EGFP and mCherry-Clc were infected with DiD-labelled IAVs and imaged with TIRF microscopy to determine the colocalization of IAVs with CCSs containing epsin without ENTH domain (Figure 2a–c). The mutant epsin constructs have comparable expression level to epsin-EGFP expressing cells (Appendix A). While we found that IAVs colocalized almost 100% with epsin-positive CCSs in RPE cells overexpressing epsin-EGFP (Figure 1b), IAVs had reduced colocalization with CCSs containing epsin ΔENTH EGFP compared to CCSs containing epsin-EGFP (Figure 2d). Similarly, IAVs also show reduced colocalization with epsin mut-H_0_ positive CCSs (Figure 2d). These results point to the reduction in strength of recruitment of epsin to CCSs containing IAVs upon removing the ENTH domain or specifically mutating the H_0_ helix. Epsin ΔENTH still possesses functional UIMs which mediate the biochemical interaction of IAVs with epsin. In contrast, removal of UIM sites along with ENTH domain significantly disrupted the colocalization of IAVs to CCSs (Figure 2d). This finding suggests a role of the ENTH domain in localizing IAVs to CCSs

### 3.3. Bulk Uptake of IAVs Is Disrupted by the Overexpression of Epsin Derivatives Lacking Functional ENTH Domain in RPE Cells

To examine the functional significance of epsin ENTH domain in IAV uptake, we utilized flow cytometry to study the bulk uptake of DiD-labeled IAVs by RPE cells expressing different epsin constructs. We looked at the rate of uptake over time and found that IAV uptake increased from 0 h to 1 h to 4 h as expected (Figure 3a). About 10% of RPE cells overexpressed epsin-EGFP internalized DiD-labelled IAVs after 1 h of incubation. About 25% of RPE cells overexpressing epsin-EGFP had internalized IAV at the end of 4-h incubation. To study whether overexpression of epsin increases the internalization of IAVs, RPE cells expressing endogenous level of epsin and overexpressing wild type (WT) epsin were incubated with DiD-labeled IAVs for 4 h. Contrary to what one might expect, overexpression of WT epsin did not increase the uptake of IAVs compared to RPE cells with endogenous expression (Figure 3b). Around 23% of RPE cells with endogenous expression and 22% of RPE cells with WT epsin overexpression had internalized IAVs at the end of 4 h. However, RPE cells overexpressing epsin ΔENTH EGFP showed reduced internalization of IAVs compared to cells overexpressing epsin-EGFP (Figure 3c). RPE cells with mutated H_0_ helix also showed similar reduction of internalization of IAVs compared to full-length epsin. Both results point to disruption of ENTH domain activity abrogating the ability of epsin to act as a cargo-specific adaptor for IAV entry.

### 3.4. Bulk Uptake of IAVs Is Disrupted by the Disruption of CME

To study the pathway dependance of IAV internalization, we disrupted CME using Pitstop 2 (hereafter referred to as Pitstop), a clathrin inhibitor. IAV internalization was reduced while CME was disrupted using Pitstop irrespective of the expression and type of epsin (Figure 4a,b). However, the internalization was not completely inhibited with activity of Pitstop, even though the percentage of IAV positive cells was reduced significantly (Figure 4c). This points toward the presence of alternate pathways for IAV internalization even if CME is disrupted. Interestingly, the cells overexpressing epsin mut-H_0_ internalized IAVs at a lower rate and this uptake rate was not further reduced by Pitstop, suggesting that non-CME pathways are involved in IAV uptake. This further illustrates the importance of intact epsin and importance of H_0_ helix in mediating internalization of IAV.

### 3.5. Visualization of Cellular Entry of Filament-Forming IAVs Using Lattice Light Sheet Microscopy

CME is the most well-characterized pathway for IAVs to gain cellular entry. Clathrin-coated vesicles canonically have spherical morphology [37]. Even though the canonical shape of a CCS is spherical, non-spherical cargos are internalized via CME [38]. We speculate that the shape of the cargo particle may play an important role in the mode of endocytic pathway utilized for cellular entry. Filamentous IAVs have been shown to utilize macropinocytosis as a mode of cellular entry [26]. Thus, we investigated whether epsin plays any role in the internalization of filament-forming IAVs. We generated filamentous IAVs (WSN-UdM) by replacing the M segment of spherical virus producing IAV strain A/WSN/1933 (H1N1) with the M segment of A/udorn/1972 (H3N2) strain. It is important to note that this strain is filament-formation-competent (hereafter referred to as filament-forming). The formation of IAV filaments was confirmed in MDCK cells infected with WSN-UdM by non-permeabilized immunostaining against influenza surface protein HA (Appendix A). The infectivity of DiD-labelled IAVs was confirmed by plaque assay (Appendix A). However, DiD-labeled WSN-UdM particles did not show filamentous nature under TIRF microscopy (TIRFM) (Appendix A). This may be due to the possibility that the majority of individual rod-shaped IAVs are around 100-300 nm in size [29] and hence the unevenness of their morphology is below optical resolution limit by TIRFM.

CME is often visualized in live cells using TIRFM. However, visualizing viral entry via CME in live cells using TIRFM is not ideal as most entry events occur at the apical side of the cell. Hence, we utilized lattice light-sheet microscopy to track the IAV entry into cells via CME. As lattice light sheet microscopy performs volumetric imaging of cells, it can visualize the entire cell surface enabling simultaneous imaging of IAV internalization in apical and basal sections of the cells (Figure 5a). From the volumetric movies of cells, 3-dimensional tracking was performed using TrackMate, a Fiji plugin [32], to detect and track IAVs binding to the cell surface followed by internalization via CME or non-CME mechanism. The time lapse montages of IAV entry in *x*-*y* and *x*-*z* spatial orientation provide information on when the IAV is internalized by the cell and whether epsin and/or clathrin is colocalized with the particular viral particle under consideration (Figure 5b). We used epsin-EGFP as a volume marker to visualize the cytoplasm near plasma membrane where IAVs bind. In *x*-*z* projection of time lapse images of tracked IAV, internalization is characterized by IAVs completely entering the illuminated cytoplasm (Figure 5b). For IAVs bound to the apical section of cells, internalization requires the *z*-position of the IAV to move towards the basal direction of the cell. Together, lattice light sheet microscopy provides an attractive alternative to conventional microscopy techniques to visualize IAV entry.

We further utilized 3D structural illumination microscopy (SIM) to visualize the colocalization and morphology of CCSs containing spherical WSN-WT and filament-forming WSN-UdM on the apical surface of the cell (Appendix A). Maximum intensity projection of 3D-SIM images of apical surface of RPE cells infected with WSN-WT and WSN-UdM for 1 h showed colocalization of both viral strains to CCSs. Both strains were recruited into CCSs containing epsin-EGFP. However, we were not able to delineate significant variations in the morphology of CCSs associated with spherical and filament-forming IAVs. We also found that WSN-UdM particles colocalized with both epsin and epsin mut-H_0_, suggesting that H_0_ helix may not play a significant role in the recruitment of epsin to WSN-UdM bound CCSs compared with WSN-WT bound CCSs.

To understand the particular difference in dynamics of internalization of spherical and filament-forming IAVs via CME, we used live cell imaging with lattice light sheet microscopy to visualize the IAV tracks. We investigated the mode of internalization of WSN-WT and WSN-UdM in RPE cells stably expressing epsin-EGFP or epsin mut-H_0_ EGFP and mCherry-Clc. It has been shown previously that filamentous viruses use macropinocytosis as the preferred mode of endocytosis for cellular entry [26]. We hypothesized that WSN-UdM particles may show lower colocalization and internalization via CCSs as they are shown to form IAV filaments. Surprisingly, time-lapse montages of IAV tracks showed both WSN-WT and WSN-UdM particles are colocalized with the CME marker, mCherry-Clc, suggesting that both filament-formation-competent and spherical-particle-only IAVs utilize CME as their entry route (Figure 5a,b).

Around 75% of WSN-WT particles and WSN-UdM particles were bound to CCSs during cellular entry (Figure 6a). Further, WSN-UdM particle-containing CCSs showed strong recruitment of epsin and epsin mut-H_0_ compared to WSN-WT bound CCSs. We quantified the strength of recruitment of epsin and epsin mut-H_0_ to WSN-WT bound CCSs and WSN-UdM bound CCSs by quantifying the maximum ratio of intensity in the puncta to background from individual IAV tracks. WSN-WT bound CCSs showed high strength of recruitment for epsin-EGFP compared to epsin mut-H_0_ (Figure 6b). WSN-UdM bound CCSs also showed reduction in strength of recruitment for epsin mut-H_0_ but to a lesser extent compared to full-length epsin. These findings agreed with the 3D-SIM images of WSN-UdM showing colocalization with both epsin and epsin mut-H_0_ (Appendix A). These findings also suggest that interference in H_0_ helix formation in epsin does not majorly impact the recruitment of epsin to WSN-UdM bound CCSs.

Mutations in H_0_ helix of epsin was shown to reduce the bulk uptake of spherical IAVs. Using flow cytometry-based IAV uptake experiments, we investigated whether mutations in epsin mut-H_0_ impact the bulk uptake filament-forming IAVs. Filament-forming WSN-UdM viruses show higher uptake than WSN-WT viruses that form only spherical particles (Figure 7a). An isogenic clone of WSN-UdM which cannot form filaments due to a point mutation (WSN-UdM1A) also showed higher uptake than that of WSN-WT. Comparing the uptake of these particles in RPE cells expressing full-length epsin and epsin mut-H_0_, we observed that uptake of WSN-WT was reduced in RPE cells expressing epsin mut-H_0_ (Figure 7a,b). However, WSN-UdM did not show statistically significant reduction in internalization in RPE cells expressing epsin mut-H_0_ compared to cells expressing full-length epsin. However, WSN-UdM1A, the isogenic clone of WSN-UdM, showed reduction in internalization to RPE cells expressing epsin mut-H_0_ compared to cells expressing full-length epsin. Together, these data confirm that filament-forming IAV recruitment is not inhibited by interference in activity of H_0_ helix in the N-terminus of epsin.

## 4. Discussion

Influenza A viruses hijack the activity of endocytic machinery involved in CME to gain cellular entry. Epsin 1, mediated by its UIMs, acts as a cargo-specific adaptor for CME of IAVs. Recent works have established the mechanical regulation of CME by epsin and other membrane-bending proteins at high membrane tension conditions [22,39]. Here, we uncovered a similar biomechanical role of epsin mediated by its H_0_ helix in IAV entry via CME.

UIMs mediate the interaction of epsin with ubiquitinated receptors at IAV binding sites on the plasma membrane [21]. We considered CALM, another membrane-bending protein from the ENTH/ANTH family which is structurally similar to epsin [36], since CALM does not possess UIMs. Recent work has shown that CALM supports clathrin-coated vesicle completion upon membrane tension increase [39]. In our experiments, CALM showed lower strength of recruitment to IAV bound CCSs compared to epsin, consistent with the role of UIMs in IAV-mediated recruitment of epsin to CCSs. Interestingly, overexpression of epsin in RPE cells did not lead to an increase in internalization of IAVs. This is consistent with earlier findings that knocking down epsin in cells did not affect the rate of internalization of IAVs [21]. Together, these findings suggest that the amount of epsin is not rate-limiting for viral entry via other endocytosis mechanisms or membrane fusion [40].

Our findings along with results from other groups showed that ENTH domain of epsin is biomechanically active with N-terminus H_0_ helix in the ENTH domain of epsin acting as a membrane tension and curvature detector [11,23,41]. Further, we showed previously that there exist complementary mechanisms mediated by ENTH domain and C-terminus intrinsically disordered domain of epsin to drive and stabilize membrane curvature necessary for the formation of CCSs at high membrane tension. Internalization of IAVs via CME involves membrane wrapping around IAVs which is energetically costly [42]. IAV tracks obtained using lattice light sheet imaging showed longer lifetimes for CCSs bound with IAVs, consistent with the earlier findings showing that IAV bound CCSs take a longer time to internalize. Motivated by these findings and given that epsin plays an important role in IAV entry via CME, we hypothesize that the biomechanical activity of epsin in detecting and driving membrane curvature is necessary for the CCS formation around IAVs. We showed that the removal of the ENTH domain from epsin reduces its strength of recruitment to IAV-containing CCSs. This result drew comparisons to our earlier findings showing the reduced nucleation of epsin without ENTH into CCSs at high membrane tension [22]. Mutation of the H_0_ helix in ENTH domain also elicited a similar reduction in the strength of recruitment of epsin to IAV containing CCSs. Further, cells overexpressing either epsin mutants without the ENTH domain or with mutated H_0_ helix showed reduction in bulk internalization of IAVs compared to cells expressing full-length epsin. This finding is contrary to the earlier results showing that bulk internalization of IAVs is not affected by the expression level of epsin. A potential explanation is that overexpression of epsin mutants with a faulty H_0_ helix may compete with and overwhelm endogenous epsin and other membrane-bending proteins from binding to IAV bound regions on the cell membrane. The overexpressed epsin mutants still possesses intact UIMs and can bind to the ubiquitinated surface receptors at IAV binding sites without initiating H_0_ helix-mediated membrane curvature generation and subsequent nucleation of CCSs around IAVs. Further, IAV uptake experiments using epsin mutant without ENTH domain and UIMs sites will be necessary to test this hypothesis.

Although studies regarding the mechanisms of IAV entry have been elegantly conducted, it remains unclear how spherical and filamentous virions utilize different endocytic routes for their entry. It has been reported that more than half of the internalized influenza viruses are associated with CCPs, while the remaining ones were found to utilize a clathrin-independent, serum-dependent pathway for their entry into cells [27,34]. Evidence showed both that filamentous IAVs can use macropinocytosis for their entry into cells, while filamentous IAVs showed a higher tendency to use macropinocytosis as their primary endocytic pathway [26]. Our study shows that filamentous IAVs may utilize both clathrin dependent and independent endocytosis for their entry. Furthermore, an increasing number of reports suggest that the shape of virions is a critical determinant for how viruses enter the cell [26,27,43]. Even though the canonical shape of clathrin-coat assembly is spherical, studies have shown that non-spherical cargo may also gain cellular entry via CME [44,45]. The internalization of vesicular stomatitis virus via CME is mediated by actin assembly and is regulated by the length of the particle [45]. These rod-shaped viral particles enter the cells through incompletely coated vesicles that internalize with the aid of actin assembly [44]. This is analogous to the internalization of CCPs at high membrane tension which is also supported by actin assembly [46]. In addition, it has been reported that rod-shaped bacteria, Listeria, hijack clathrin-dependent endocytosis for their entry [47]. Together with our findings, filamentous IAV and bacteria may share similar entry mechanisms which are dependent on their shape.

To investigate the pathways through which filamentous IAVs enter cells, we generated mutant IAVs by replacing the M segment of a spherical laboratory strain with the M segment of an IAV strain that can produce filamentous particles. Individual particles produced by filament-forming IAVs were not distinguishable from spherical IAVs under TIRF microscopy; however, upon infecting MDCK cells they formed clearly visible filaments, some of which were micrometers long. Compared to spherical IAVs, filament-forming IAVs showed increased bulk internalization into RPE cells. One caveat is that we added the same number of infectious particles, but this does not necessarily mean the same number of physical particles. However, our DiD-labeled particles showed that the number of particles was similar between different conditions (Appendix A).

Both 3D-SIM and live-cell tracks of filament-forming IAVs showed that these particles are strongly recruited to and are internalized via CCSs. Further, the strength of recruitment of epsin to filament-forming IAVs did not show a drastic reduction as seen in the case of spherical IAVs when the epsin mutant with compromised H_0_ helix was exogenously expressed. In addition, there was no significant drop in bulk internalization of filament-forming IAVs to RPE cells overexpressing epsin mut-H_0_ as seen in spherical IAVs. Furthermore, the isogenic clone of filament-forming IAV strain with a point mutation preventing the formation of IAV filaments, showed similar uptake characteristics as that of spherical IAVs. Thus, the unique internalization characteristics of filament-forming IAVs were likely mediated by the shape of the particles or the number of HA, NA (neuraminidase) or M2 molecules per particle. Further investigations are required to uncover how they enable their entry without the membrane curvature-generating activity of H_0_ helix. We hypothesize that membrane curvature generation by epsin may not play a significant role in the internalization of filament-forming IAVs via CME. Perhaps activity of actin cytoskeletal assembly is responsible for their entry via CME similar to the case of filamentous stomatitis virus. Together, our investigation into IAV entry via CME suggests a biomechanical role of epsin in enabling viral entry of spherical IAVs. Although filament-formation-competent IAVs strongly recruit epsin, they did not appear to require the membrane curvature detection and generation of epsin mediated by its H_0_ helix.

## Figures and Tables

**Figure 1 membranes-12-00859-f001:**
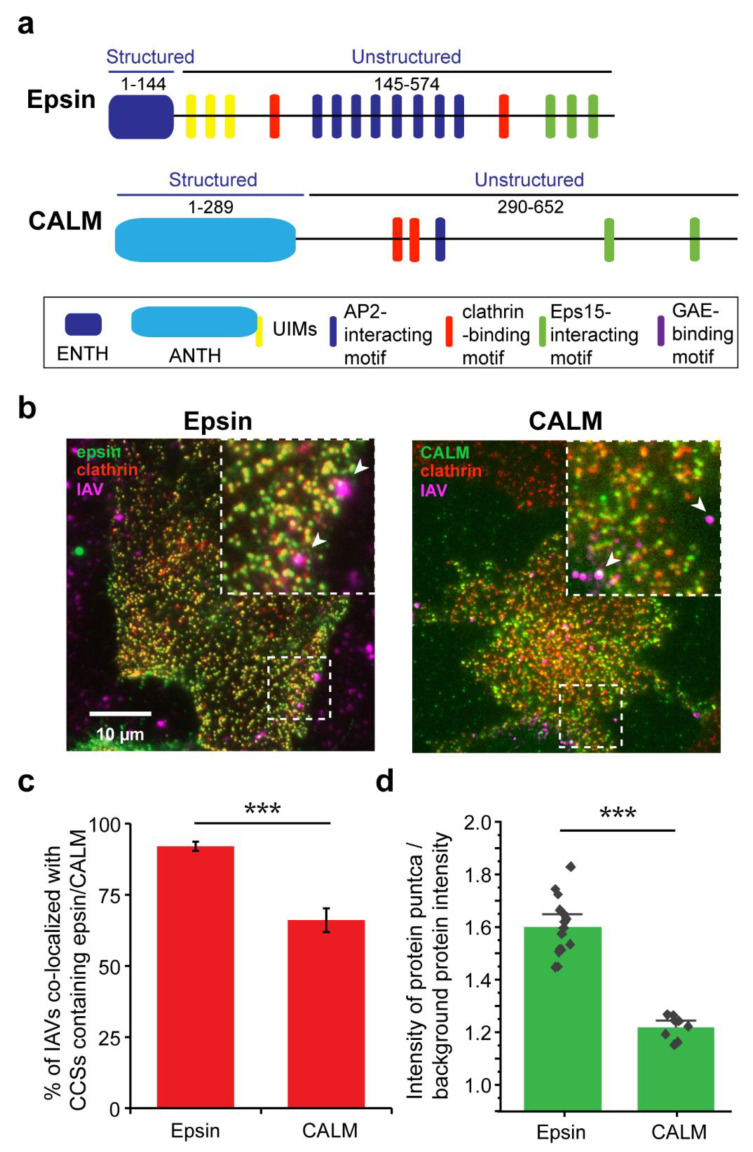
Colocalization of IAVs to ENTH/ANTH proteins containing CCSs. (**a**)**.** Domain structures of epsin and CALM. GAE: gamma-adaptin ear. (**b**) IAVs bound to surface of RPE cells overexpressing epsin-EGFP and mCherry-Clc (left), or CALM-mCherry and EGFP-Clc (right, pseudo colored). White arrows in inset shows IAVs bound to cell surface. (**c**) Percentage of IAVs colocalizing with epsin and CALM to the total number of IAVs bound to the surface. The error bars denote counting error from binomial distribution. (**d**) Ratio of puncta intensity to background intensity of proteins colocalized with surface bound IAVs. For c and d, N_cells_ expressing epsin-EGFP were 15 and N_cells_ expressing CALM-mCherry were 10. For c and d, N_cells_ expressing epsin-EGFP was 15, and N_cells_ expressing CALM-mCherry was 10. N_IAVS_ analyzed for epsin-EGFP and CALM was 288 and 128 respectively. For c the error bars denote standard error from binomial distribution. For d the error bar denotes standard error. *** represents *p* < 0.001.

**Figure 2 membranes-12-00859-f002:**
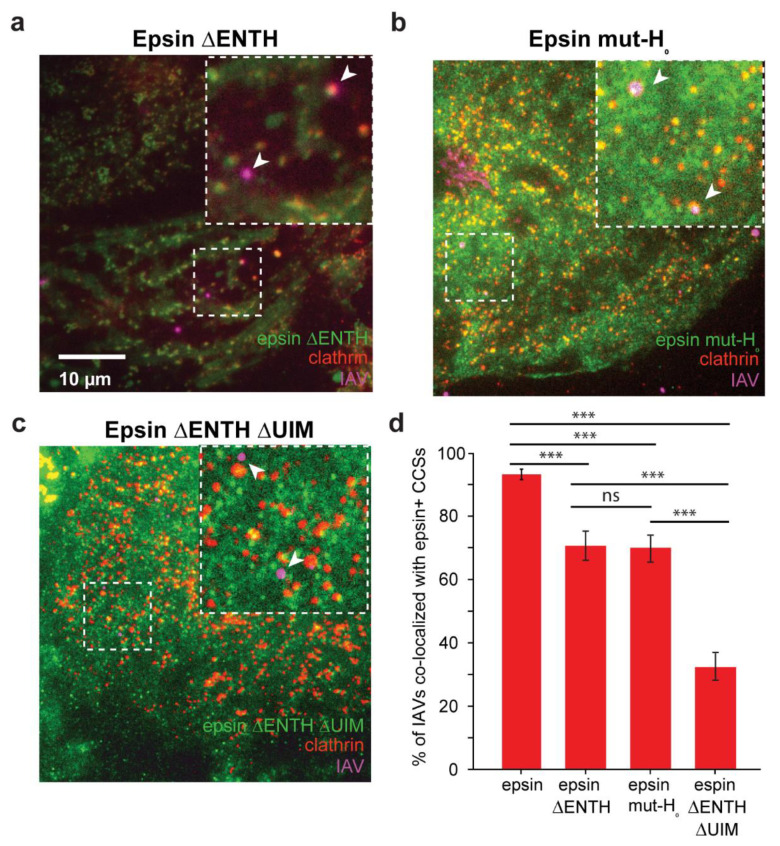
Colocalization to IAVs to CCSs is disrupted in cells overexpressing epsin without a functioning ENTH domain. (**a**–**c**) IAVs bound to surface of RPE cells stably expressing epsin ΔENTH EGFP, epsin mut-H_0_ EGFP, epsin ΔENTH ΔUIM EGFP and mCherry Clc, respectively. White arrows in inset show IAVs bound to the cell surface. (d) Fraction of IAVs colocalized with CCSs containing epsin mutants. For (**d**), N_cells_ expressing epsin-EGFP, epsin ΔENTH EGFP, epsin mut-H_0_ EGFP, epsin ΔENTH ΔUIM EGFP were 15, 15, 15, 15, respectively. N_IAVS_ analyzed for epsin-EGFP, epsin ΔENTH EGFP, epsin mut-H_0_ EGFP, epsin ΔENTH ΔUIM EGFP were 288, 120, 147, 129, respectively. For b the error bars denote standard error from binomial distribution. The data for epsin-EGFP is reproduced from Figure 1c. The error bars denote counting error from binomial distribution. ns and *** represent not significant and *p* < 0.001 respectively.

**Figure 3 membranes-12-00859-f003:**
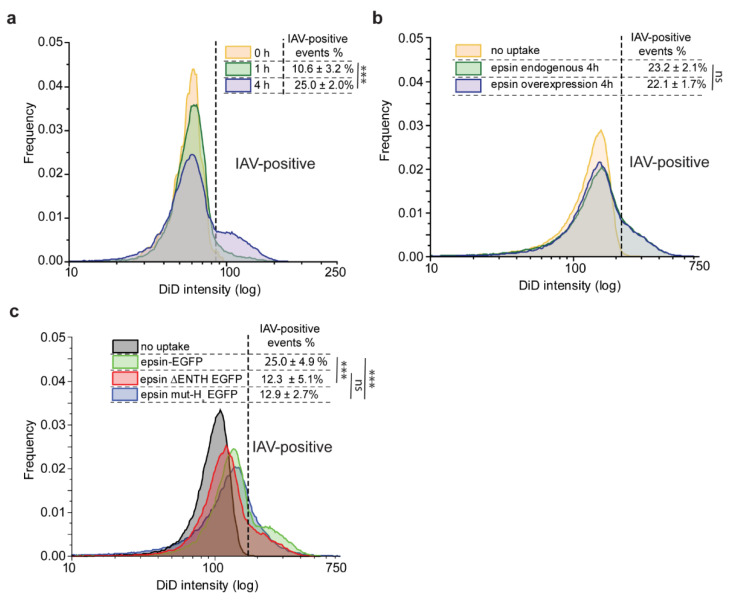
Internalization of IAVs is disrupted in cells overexpressing epsin without functioning ENTH domain. (**a**) Flow cytometry histograms showing IAV uptake at 0 h (no infection), 1 h and 4 h in RPE cells overexpressing epsin-EGFP. (**b**) IAV uptake (4 h) in RPE cells with endogenous expression of epsin and overexpression of epsin WT. (**c**) IAV uptake (4 h) in RPE cells stably expressing epsin-EGFP, epsin ΔENTH EGFP and epsin mut-H_0_ EGFP. Percentage of cells with IAV internalization is shown in the inset. Experiments were repeated n = 3 and standard deviation is provided in the inset. ns represents not significant. *** represent *p* < 0.001.

**Figure 4 membranes-12-00859-f004:**
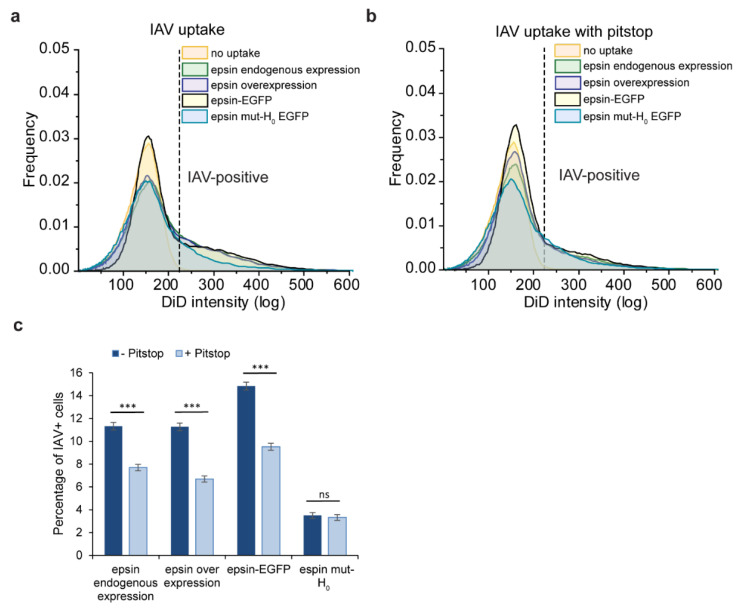
Internalization of IAVs is disrupted by inhibition of clathrin-mediated endocytosis. (**a**) IAV uptake (4 h) in RPE cells with endogenous expression of epsin and overexpression of epsin WT, cells stably overexpressing epsin-EGFP and epsin mut-H_0_ EGFP. (**b**) IAV uptake (4 h) in RPE cells with endogenous expression of epsin and overexpression of epsin WT, cells stably overexpressing epsin-EGFP and epsin mut-H_0_ EGFP pretreated with Pitstop. (**c**) Percentage of cells with IAV internalization with and without Pitstop treatment. Experiments were repeated n = 3. ns represents not significant. *** represent *p* < 0.001.

**Figure 5 membranes-12-00859-f005:**
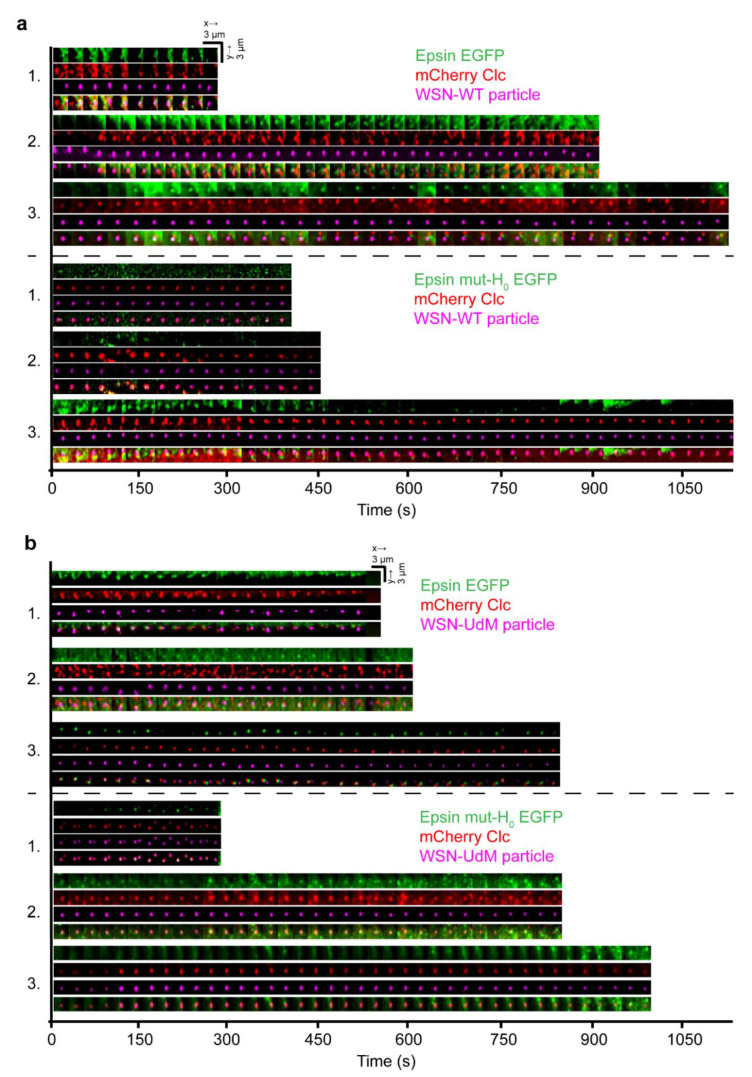
Both spherical and filament-forming IAVs co-localize and internalize via CCSs. (**a**) Three example trajectories of IAV (magenta) internalization in x-y and x-z spatial orientations. Time lapse montages of WSN-WT particles co-localizing with epsin-EGFP and mCherry-Clc (top panel) and epsin mut-H_0_ EGFP and mCherry-Clc (bottom panel). (**b**) Time lapse montages of WSN-UdM particles co-localizing with epsin-EGFP and mCherry-Clc (top panel) and epsin mut-H_0_ EGFP and mCherry-Clc (bottom panel).

**Figure 6 membranes-12-00859-f006:**
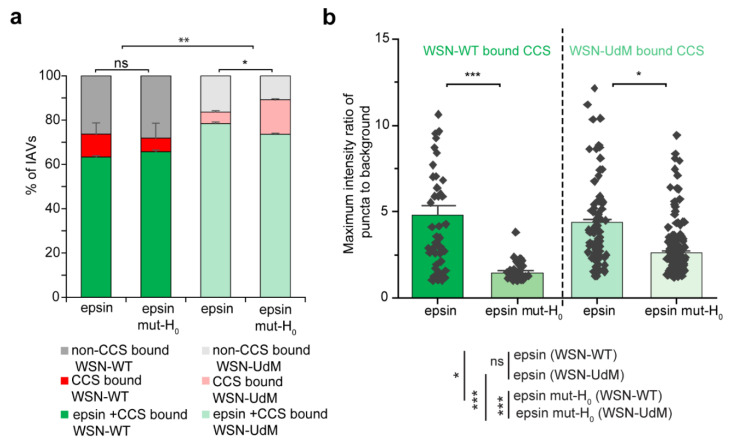
CCSs containing filament-forming IAVs strongly recruit epsin. (**a**) Percentage of WSN-WT and WSN-UdM particles bound to CCSs colocalizing with or without epsin or epsin mut-H_0_. (**b**) Maximum ratio of puncta intensity to background intensity of epsin or epsin mut-H_0_ colocalized with IAV-containing CCSs. For a and b, N_IAV_ tracks for WSN-WT in cells expressing epsin-EGFP, WSN-WT in cells expressing epsin mut-H_0_ EGFP, WSN-UdM in cells expressing epsin-EGFP and WSN-UdM were 91, 105, 180, 225, respectively, and corresponding N_cells_ were 18, 14, 15, 21 respectively. The error bars denote standard error. ns represent not significant. *, **, and *** represent *p* < 0.05, *p* < 0.01 and *p* < 0.001 respectively.

**Figure 7 membranes-12-00859-f007:**
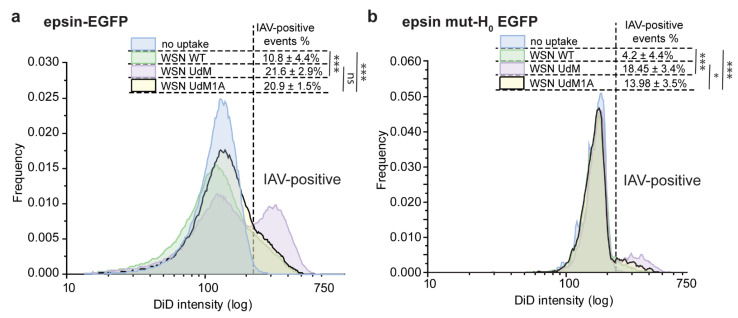
Bulk uptake of WSN-UdM is not affected by mutation H_0_ in epsin. (**a**) Uptake (4 h) of WSN-WT, WSN-UdM and WSN-UdM1A by RPE cells stably expressing epsin-EGFP. Percentage of cells with IAV internalization is shown in the inset. (**b**) Uptake (4 h) of WSN-WT, WSN-UdM and WSN-UdM1A by RPE cells stably expressing epsin mut-H_0_ EGFP. Percentage of cells with IAV internalization is shown in the inset. For a and b, experiments were repeated n = 3 and standard deviation is provided in the inset. ns represents not significant. *, and *** represent *p* < 0.05 and *p* < 0.001 respectively.

## Data Availability

The data presented in this study are available on request from the corresponding author.

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
