# Peer review of "Biomechanical Role of Epsin in Influenza A Virus Entry"

_membranes, 2022, doi:10.3390/membranes12090859_

Round 1

Reviewer 1 Report

The authors studied how epsin is involved in the IAVs endocytosis through CME and CCSs using different principles of live  cells and optical microscopies and flow cytometry. The study concluded the relative role of epsin in IAVs internalization both in their spheric and filaments forms. The experimental designs and results are very well conducted and conclusions open to new hypothesis about the biomechanisms of IAVs to entry into cells.

Minor comments.

Line 122 to 124: An statistical analysis is required to compare the colocalization of IAVs with CCSs in cells transfected with ΔENTH EGFP vs cell containing epsin-EGFP. It is not valid to compare figures represented in Figure 2b & Figure 1b. 

Reviewer 2 Report

Internalisation of influenza A virus particles is mediated by various endocytosis pathways. Internalisation by clathrin-mediated endocytosis is a common pathway and requires receptor binding and adaptor proteins. One of these adaptor proteins is Epsin-1; which binds membranes, clathrin, ubiquitinated cargo and additional accessory proteins. In its N-terminus Epsin-1 has an amphipathic helix,  Helix zero, that forms upon binding to lipids and inserts into the membrane bilayer and induces curvature. The authors hypothesise that for a cell to internalise the influenza A virus (IAV) membrane bending is initiated by the virus, epsin recruited where helix zero induces further membrane curvature leading to internalisation.  They test this hypothesis by using wild type and mutants of Epsin-1 lacking helix zero to analyse and compare co-localisation with IAV and virus uptake. The results demonstrate that cells expressing epsin-1 lacking helix zero have a reduced uptake of IAV but does not appear to negatively affect the uptake of filamentous IAV.

Comments:-

1. Please include a more descriptive definition of epsin mut-H0 in your methods section (line 394). You write that helix zero was substituted to histidine. Did you delete the 18 amino acids that constitute helix zero and insert a histidine?  Did you use the L6H mutation of helix zero that has been shown to reduce the lipid association of the epsin-1 ENTH domain and thereby reduce its recruitment to membranes (Ford et al., Nature 2002)?

2. The quantification of the data generally has a high quality. Please be consistent and include the number of cells/images/experimental repeats analysed for each graph. For example Figure 1d contains information on sample size but Figure 1c does not.

3. Statistical analysis. Please include a subheading in the methods section on statistical analysis. Which statistical tests have been used to analyse the data?

4. Figure 3 - How many cells were analysed per experimental repeat? Is there a statistical significant difference in the inhibition of internalisation of IAVs in cells overexpressing wt epsin1, epsin1-deltaENTH or epsin1 mut-H0? This information would further solidify the data and the conclusion.  

5. The quantification in Figure 6b shows a strong recruitment of epsin mut-H0 to filamentous IAVs. It would be useful to see examples of images used for this quantification in the figure. The three examples of co-localisation between WSN-UdM and epsin mut-H0 in Figure 5b does not show a prominent co-localisation. The background is high in these images.  What is known from the literature is that helix zero of epsin1 is important for membrane recruitment and curvature induction of the protein in clathrin-mediated endocytosis. For IAV internalisation what is the relative importance of the epsin UIM versus the ENTH-H0 for recruitment?  

6. Figure 7 shows a reduced uptake of filamentous IAVs in cells expressing epsin mut-H0. Please include a statistical analysis to strengthen this conclusion. 

7. Please comment in the discussion on whether filamentous IAVs are more likely to use non-clathrin/epsin-dependent endocytosis. Please discuss potential similarities of filamentous IAVs and bacterial entry (Veiga & Cossart, Nature Cell Biol, 2005; etc.)  through clathrin/epsin-assisted endocytosis. 

8. Please check the formatting of the bibliography.  For example reference 23 is not formatted correctly (last author is listed first).

Reviewer 3 Report

The work by Joseph et al. examined the role of epsin in clathrin-mediated endocytosis of influenza A virus.  This work combined single virus imaging and co-localization with clathrin mediated machinery as well up flow cytometry based assays to determine amount of influenza entering the cells when different epsin constructs were expressed.  This paper elegantly teased out that influenza A required the Ho helix of epsin to recruit epsin to clathrin coated pits.  Without this function of epsin influenza uptake was greatly reduced.  Interestingly, using filamentous influenza - this requirement for epsin was lost and deletion of the Ho helix no longer effected virus uptake.  This is reminiscent to studies (an in strong agreement with them) by Cureton et al. that showed the size and shape of VSV dictated the dynamics of clathrin mediated endocytosis of that virus. 

I strongly endorse publication of this work.  I have one major point that would be nice if the authors could addressed and some minor points that should be left to the discretion of the authors.

Major point:

1. There is no control for if the labeled virus is still infectious.  When too much dye is used in virus labeling, it has been shown to inhibit membrane fusion (but not particle uptake).  Could a simple infectivity assay +/- the dye labeling be performed.  It is reasonable and expect the particles should lose some infectivity (event 20-50% is acceptable) but it should at least be confirmed that the virus is not 100% dead.

Minor points:

1. You are heavily relying on over expression of proteins.  What are the concerns that the unlabeled endogenous protein is still playing a role in influenza entry when you express the mutations/ truncations of proteins?

2. All your imaging for co-localization was by TIRFM.  I am aware that some virus does get between the cell and the glass but the largest portion of virus being would be on the top of the cell.  How do you know the dynamics of the clathrin mediated entry and molecule requirements for endocytosis do not change on the face of the cell not bound and coupled to the glass?  Could the membrane tension and bending on the top side of the cell be different and could the change the results?

3. Does the loss of the Ho helix of epsin decrease the amount of infection in these cells?

4.  To inhibit CME the drug Pitstop was used. There are a lot of off target effects with Pitstop and usually cells look extremely unhealthy after Pitstop treatment.  Have you confirmed the results of this drug any other way (dominate neg mutation of dynamin? dynasore? or there are new specific peptide inhibitors of clathrin from Sandy Schmid's lab called Wbox2 which does not have the off target effects).  
